# Defect Coverage after Forequarter Amputation—A Systematic Review Assessing Different Surgical Approaches

**DOI:** 10.3390/jpm12040560

**Published:** 2022-04-01

**Authors:** Denis Ehrl, Nikolaus Wachtel, David Braig, Constanze Kuhlmann, Hans Roland Dürr, Christian P. Schneider, Riccardo E. Giunta

**Affiliations:** 1Department of Hand, Plastic and Aesthetic Surgery, University Hospital, LMU Munich, Marchioninistraße 15, 81377 Munich, Germany; denis.ehrl@med.uni-muenchen.de (D.E.); david.braig@med.uni-muenchen.de (D.B.); constanze.kuhlmann@med.uni-muenchen.de (C.K.); riccardo.giunta@med.uni-muenchen.de (R.E.G.); 2Orthopaedic Oncology, Department of Orthopaedics and Trauma Surgery, University Hospital, LMU Munich, Marchioninistraße 15, 81377 Munich, Germany; hans_roland.duerr@med.uni-muenchen.de; 3Department of Thoracic Surgery, University Hospital, LMU Munich, Marchioninistraße 15, 81377 Munich, Germany; christian.schneider@med.uni-muenchen.de

**Keywords:** forequarter amputation, targeted muscle reinnervation, osteomusculocutaneous flap, fillet flap, epaulette flap, interscapulothoracic amputation, spare parts, microsurgery, reconstructive surgery

## Abstract

Autologous fillet flaps are a common reconstructive option for large defects after forequarter amputation (FQA) due to advanced local malignancy or trauma. The inclusion of osseous structures into these has several advantages. This article therefore systematically reviews reconstructive options after FQA, using osteomusculocutaneous fillet flaps, with emphasis on personalized surgical technique and outcome. Additionally, we report on a case with an alternative surgical technique, which included targeted muscle reinnervation (TMR) of the flap. Our literature search was conducted in the PubMed and Cochrane databases. Studies that were identified were thoroughly scrutinized with regard to relevance, resulting in the inclusion of four studies (10 cases). FQA was predominantly a consequence of local malignancy. For vascular supply, the brachial artery was predominantly anastomosed to the subclavian artery and the brachial or cephalic vein to the subclavian or external jugular vein. Furthermore, we report on a case of a large osteosarcoma of the humerus. Extended FQA required the use of the forearm for defect coverage and shoulder contour reconstruction. Moreover, we performed TMR. Follow-up showed a satisfactory result and no phantom limb pain. In case of the need for free flap reconstruction after FQA, this review demonstrates the safety and advantage of osteomusculocutaneous fillet flaps. If the inclusion of the elbow joint into the flap is not possible, we recommend the use of the forearm, as described. Additionally, we advocate for the additional implementation of TMR, as it can be performed quickly and is likely to reduce phantom limb and neuroma pain.

## 1. Introduction

The multimodal treatment of primary malignant bone or soft tissue tumors involves multiagent chemotherapy, radiotherapy, and wide surgical resection. Since the combination and enhancement of these regimes, long-term survival rates have improved significantly over the last decades and the operative treatment advanced from direct amputation to limb-sparing surgery [1]. However, for patients with locally advanced tumors of the limbs, amputation remains the only curative option.

With regard to the upper extremity, interscapulothoracic amputation (ISTA) or forequarter amputation (FQA) is the most radical ablative procedure [2]. It involves the amputation of the complete upper extremity, including the anatomic structures of the shoulder girdle, leading to the loss of the shoulder silhouette. Besides malignant bone tumors, post-radiation defects and traumatic scapulothoracic dissociation are the most common indications for this rare procedure [2,3,4]. Depending on the extent of FQA, direct wound closure is often not possible. A common reconstructive option for large defects after FQA are fillet flaps. These are harvested from the amputated limb and do not create any additional donor side morbidity. Depending on the surgical technique, different flap designs have been established for the upper extremity—the fasciocutaneous, the musculocutaneous, and the osteomusculocutaneous fillet flap [5,6,7]. While the first two techniques are viable options for the coverage of skin and soft-tissue defects, the inclusion of osseous structures can also stabilize the thoracic wall and reconstruct the shoulder contour. Hence, this flap is called the “epaulette” flap, similar to the ornamental shoulder piece of military uniforms [2]. Besides cosmetic advantages, the “epaulette” flap also creates a stable osseous and soft-tissue envelope that provides a socket for an upper-limb prosthesis [7,8].

Recent developments in prosthetic medicine have created devices that offer multi-functional joints with fine motor capabilities, as well as improved comfort and aesthetics [9]. With the aim to accelerate the cortical control of these advanced prosthetic systems, the concept of targeted muscle reinnervation (TMR) was presented in 2002. In TMR, transected peripheral nerves are transferred to recipient motor nerves of residual muscles in the amputated limb in order to avert muscle atrophy and reinitiate organized muscle innervation [10,11]. Prior studies revealed that TMR additionally significantly reduces the risk of developing neuromas and phantom limb pain [9,12,13]. These two characteristics make TMR a viable option for reconstructive procedures following curative or palliative FQA.

The aim of this article was to systematically review previous approaches for chest wall stabilization and shoulder contour reconstruction after FQA with osteomusculocutaneous free fillet flaps. Moreover, we report on a case with an alternative surgical technique that incorporates the forearm, wrist, and metacarpus of the amputated limb into the flap in combination with TMR. The simultaneous utilization of TMR provides the first results on feasibility and improved neuropathic pain management after FQA and osseous “spare-parts” reconstruction.

## 2. Methods

The identification of studies for this review was based on the Preferred Reporting Systems for Systematic Reviews and Meta-Analysis (PRISMA) statement [14]. The MeSH terms “fillet flap”, “epaulette flap”, “osteomusculocutaneous flap”, “forequarter amputation”, and “interscapulothoracic amputation” were used to search the PubMed and Cochrane databases for publications that focus on shoulder contour reconstruction and thoracic wall stabilization with osteomusculocutaneous free fillet flaps after FQA. The literature search was completed on 3 February 2022. In total, 387 articles were identified (Figure 1). All of the results were imported into the Covidence systematic review software (Veritas Health Innovation, Melbourne, Australia) for the removal of duplicates.

First, the titles and abstracts of the citations were individually scrutinized to determine which were relevant to the review. Subsequently, studies were excluded if they were not available in English or German, or if they described the usage of cutaneous and musculocutaneous (soft-tissue) fillet flaps or other reconstructive procedures (non-fillet flaps) after FQA. This resulted in the identification of six articles that describe the usage of osteomusculocutaneous fillet flaps for reconstruction following FQA. Two of these were excluded from the study as they did not provide enough details on the surgical technique used, as well as outcome measurements [7,15]. Four studies remained that met the inclusion criteria.

## 3. Results

### 3.1. Literature Search

Ten cases of shoulder contour reconstruction and thoracic wall stabilization with osteomusculocutaneous free fillet flaps after FQA were described in the four publications that were identified (Table 1). In 8 out of 10 cases, FQA was a consequence of a local malignancy, whereas two cases were due to prior trauma. Four patients required palliative FQA, the rest were treated in a curative intention (*n* = 6). All of the patients survived the initial surgery.

Considering the vascular supply of the flap, the subclavian artery was predominantly anastomosed to the brachial artery (*n* = 8). Other options were the suprascapular artery (*n* = 1) and the internal thoracic artery (*n* = 1). In 5 out of 10 cases, two venous anastomoses were completed. In the majority of cases, the brachial or cephalic vein of the flap was connected to the subclavian or the external jugular vein (*n* = 7). Three cases required revision surgery due to an infected hematoma (*n* = 1), partial skin necrosis (*n* = 2), and/or arterial thrombosis (*n* = 1). Another case required tumor re-excision after the confirmation of a R1 tumor margin. Three out of a total of six sarcoma patients developed pulmonary metastases within the first 18 months after surgery, and two of these patients died within the observation period. Another patient died 13 years after the initial operation from a local recurrence of the sarcoma. Both patients with traumatic FQA survived the follow-up period.

### 3.2. Case Report

A 25-year-old male patient with a large chondroblastic osteosarcoma of the left upper humerus presented to our hospital and was treated by our multidisciplinary sarcoma team (Table 2). The patient was initially diagnosed eight months earlier but unfortunately decided to pursue an alternative treatment with a homeopathic practitioner for three months. When he returned to our hospital, the tumor had increased considerably in size and was staged as cT2 cN0, cM1, infiltrating the glenohumeral joint, the muscles of the upper arm and rotator cuff, as well as the latissimus dorsi and both pectoral muscles (Figure 2). Additionally, three suspect pulmonary lesions, thrombosis of the subclavian, axillar and brachial vein was observed on CT-Angio scans. Due to oligometastasis and the young age of the patient a curative treatment approach was initiated, including neoadjuvant chemotherapy followed by wide tumor resection. Oncological treatment following the protocol of the European and American Osteosarcoma Study Group (EURAMOS-1) was initiated [19,20]. After 6 weeks of chemotherapy, wide resection was conducted. The surgery was performed in an interdisciplinary collaboration of Tumor Orthopedics, Thoracic and Plastic Surgeons. Due to the extent of the sarcoma, an extended FQA, including an atypical lung segment resection and the resection of the first three ribs as well as the complete clavicle, was necessary to allow complete resection. Suspect pulmonary lesions were individually resected by atypical lung resections.

Tumor infiltration into the distal end of the humerus prohibited a reconstructive option that used the elbow joint [17,18]. We therefore decided to use an osteomusculocutaneous free fillet flap from the tumor-free forearm for defect coverage and shoulder contour reconstruction. The flexed wrist and metacarpal bones were incorporated into the flap to create a shoulder contour that would function as a prosthetic socket. The radius and ulna were shortened at the proximal end to a length of 14 cm and the skin and soft tissue were opened longitudinally on the ventral aspect of the forearm. The radius was then attached to the sternum with a plate. Subsequently, microsurgical anastomoses were performed between the brachial artery and thoracoacromial artery and between the cephalic vein and remaining stump of the subclavian vein in end-to-end technique with interrupted sutures (total time of ischemia: 165 min). The metacarpal bones were connected to the lateral thoracic wall using a non-resorbable suture to ensure lateral stability of the construct. TMR was performed by epineural coaptation of the three forearm motor-nerves (i.e., median, radial, and ulnar nerve) to the three trunks of the brachial plexus (Figure 3). The skin and soft tissue of the flap was fitted to the chest wall. The patient was transferred to the intensive-care unit after surgery and was extubated on the following day. Pathology confirmed R0-resection and complete wound healing occurred without infection or tissue loss. The patient was discharged 11 days after surgery.

Clinical follow-up showed a solid osseous framework with good protection of the thoracic organs (Figure 4) and an acceptable improvement of the shoulder contour (Figure 5). The patient did not suffer from phantom limb or neuroma pain. Moreover, neurological examination six weeks after surgery showed an increasing tactile sensation of the flap. However, pulmonary and lymph-node metastases occurred during follow-up, resulting in the continuous deterioration of the clinical condition. A palliative therapeutic concept commenced, and the patient died three months after tumor resection.

## 4. Discussion

Improved reconstructive options and a multimodal treatment of mesenchymal tumors, such as osteosarcomas, enabled limb salvage in the majority of patients [21,22]. Nevertheless, radical tumor resection is usually necessary as inadequate surgical margins are significantly associated with higher local recurrence rates and decreased overall survival of patients [15,16,23]. With regard to locally progressed primary tumors of the proximal upper extremity, limb salvage is not always possible. In these cases, FQA allows for wide resection margins [24]. However, albeit technically feasible, the radical ablation of the arm and the anatomical structures of the shoulder girdle are associated with severe comorbidities, such as possible life-threatening intraoperative hemorrhage, as well as respiratory impairment or even failure [16,25], especially when including the resection of the chest wall and/or parts of the lung [16,26]. Nevertheless, while such a radical procedure has to be carefully assessed for its advisability for each individual patient, FQA allows for wide resection margins and, thus, a curative treatment concept [24,25].

A common consequence of the radical ablation of the arm and the anatomical structures of the shoulder girdle is the requirement of a subsequent microsurgical reconstruction in order to enable adequate defect coverage and wound closure. The “spare-parts concept”, which utilizes tissue from the amputated limb to reconstruct a defect without creating additional donor side morbidity (i.e., fillet flap), is a recognized technique in reconstructive and traumatic surgery [7,15,27,28,29]. Küntscher and colleagues provided a thorough overview of this surgical technique in an extensive study on 104 fillet flaps. The authors classify fillet flaps into pedicled finger and toe, pedicled limb, and free filet flaps [15]. With regard to free fillet flaps used for reconstructive surgery after FQA, flaps have been predominantly described according to their tissue content (fasciocutaneous, musculocutaneous, and osteomusculocutaneous) [5,6]. In contrast to fasciocutaneous and musculocutaneous flaps, the inclusion of bones in the osteomusculocutaneous fillet flap enables restoration of the shoulder silhouette and provides additional stability, as well as protection of the thorax and its inner organs, when using this technique for reconstruction after FQA [5,6,7,30]. Indeed, the reconstruction of the chest wall integrity after extensive resection is of the highest priority as a reduced structural integrity of the chest is associated with paradox respiratory movement and therefore impaired ventilatory function [7,31]. Alternative methods that offer stability, such as the use of alloplastic materials (e.g., synthetic mesh), have been successfully described for the reconstruction of large chest wall defects [32,33]. These often require additional soft tissue coverage, commonly by extensive free flaps in the case of FQA. However, due to the risk of predominantly intraoperative bacterial contamination, these are associated with a higher rate of infection and impaired wound healing [34,35,36,37,38].

Instead of utilizing the amputated limb, local or free flap reconstructive options may be used for defect coverage after FQA. Indeed, several reports exist, which demonstrate the successful use of these techniques, such as the fasciocutaneous deltoid, the tensor fascia latae (TFL), or the TFL + rectus femoris flap [7,39,40]. Similarly, extensive studies have been published regarding the reconstruction of the chest wall using primarily local myocutaneous flaps [41,42,43]. However, in the case of FQA, where the subscapular system is often severed, pedicled flaps like latissimus dorsi, (para)scapular, and serratus flaps become unavailable local options. In addition, when compared to an osteomusculocutaneous fillet flap, these procedures involve additional donor site morbidity, as well as limited stability for the shoulder girdle.

Despite the potential physiological and psychological benefits for the patient, very few cases of osteomusculocutaneous free fillet flaps for chest wall and shoulder reconstruction have been described in previous publications. Our systematic literature search identified four articles that give detailed information on this technique in a total of ten cases (Table 1) [2,16,17,18]. Hitherto published surgical techniques on osteomusculocutaneous free fillet flaps after FQA can be separated into two main subgroups, depending on the choice of bones that were included into the flap—the approach by Steinau et al. as well as the one by Kuhn and colleagues, who described the utilization of the radius and/ or ulna to stabilize the thoracic wall [2,16]. Thus, Steinau et al. reported on three cases in which the proximal forearm was fixated to the remaining parts of the clavicle or the sternum with the distal ends of the radius and ulna being attached to the thoracic wall [2]. Two years later, Kuhn et al. published a case report describing the use of the bones and soft-tissue of the forearm to reconstruct the thoracic wall and to allow for mediastinal protection after extended FQA, including complete anterior and posterior chest wall resection, as well as pneumectomy [16].

In contrast, in their subsequently published articles, Osanai and colleagues as well as Koulaxouzidis et al. described techniques that connect the distal humerus to the clavicle or the sternum (in cases where a complete resection of the clavicle was necessary) and created a lateral prominence that resembles the natural silhouette of the shoulder [17,18,37,44,45]. Thus, Osanai et al. included the flexed elbow joint into the flap to imitate the natural shoulder contour through the eminence of the olecranon [17]. Similarly, Koulaxouzidis et al. also used the elbow joint for shoulder contour reconstruction in a total of four patients [18]. Moreover, this technique also involved the reconstruction of the axillary fold in addition to the shoulder contour. The authors avoided large scale soft-tissue separation to preserve the outline of the elbow, and the cubital skin crease was thus used to recreate the axilla.

The results illustrate that the functional and cosmetic outcomes are highly dependent on the location and extent of the tumor or trauma. Local tumor progression of our patient required the resection of the complete clavicle, scapula, and the first three ribs. The upper arm had to be excluded from the flap to accomplish tumor free margins, permitting the previously published surgical approach that incorporates the elbow joint into the flap [17,18].

Thus, our surgical options were limited to the utilization of the forearm and the hand, as the osteomusculocutaneous free flap for the reconstruction of the thoracic wall and shoulder. Therefore, we decided to use a new surgical approach—by connecting the radius to the sternum, we were able to utilize the flexed wrist instead of the elbow joint to imitate the natural shoulder contour (Table 2). Short-term follow-up of the patient revealed limited appearance of the shoulder silhouette; however, sufficient protection of the thoracic organs through the osseous framework of the flap, and good anatomical conditions for attachment of a socket prosthesis. Consequently, we present a third subgroup of osteomusculocutaneous fillet flaps for reconstruction with “spare-parts” after FQA that incorporates the bones of the forearm, wrist, and metacarpus. Unfortunately, we cannot provide long-term results using this technique due to the rapid development of metastatic disease, resulting in the patient’s death three months after surgery.

Common complications following extremity amputation are the occurrence of phantom limb pain and the development of neuromas [11]. The prevalence of phantom limb pain ranges between 45 and 85%, and typically displays two peaks in its incidence: one month and one year after amputation [46,47]. The concept of TMR was initially developed to enable advanced control of myoelectric prostheses, but has also shown to significantly reduce phantom pain and neuroma formation after limb amputation [13,47]. To exploit these features, we connected the trunks of the brachial plexus to the forearm nerves in the fillet flap. The epineural end-to-end coaptation required only a little extra time due to the large caliber of the peripheral nerves and was completed within 35 min. The patient did not develop any phantom limb or neuroma pain during the follow-up period. These findings provide new data on the feasibility and possible functional improvement of phantom pain management in patients who undergo fillet flap reconstruction after FQA. While the validity of our results is significantly limited due to the short observation period and single-case experience, previous studies have demonstrated the high effectiveness of TMR when performed as a preemptive measure, as well as when used as a treatment option for patients with postamputation pain [48,49,50,51]. Indeed, Mioton and colleagues demonstrated significant improvements of residual limb and phantom limb pain parameters in 33 patients with major limb amputations due to TMR one year after treatment [49]. Moreover, in a recent prospective randomized clinical trial, TMR was shown to reduce chronic pain in amputees when compared with the gold standard (excision and muscle burying) [48]. Similar results were shown by Valerio et al. when implementing TMR as a preemptive measure to reduce chronic postamputation pain [50]. In their multi-institutional cohort study, the authors demonstrated that patients who underwent TMR had less phantom and residual limb pain when compared with untreated amputee controls. This effect was shown across all subgroups. Considering these previous findings and also the outcome of our presented case, a high benefit to cost ratio of TMR along with fillet-flap reconstruction after FQA seems highly likely.

In the majority of reports, including our own case, tumor infiltration of the shoulder girdle or chest wall was the indication for FQA. Using the “spare-parts” of an extremity that was impaired by local cancerous progression asks the question of whether this procedure is a safe oncological approach. In this context, the fillet flap technique uses the identical principle of the resection–replantation technique reported by Windhager et al. [52]. The authors resected the tumor-bearing area of the upper extremity and replanted the distal part of the arm to the proximal stump. None of the 12 patients developed a local recurrence within the follow-up period. Furthermore, a different study by Ver Halen et al. described 27 soft tissue fillet flaps from the upper and lower extremity after soft tissue malignancies; none of the patients developed cancer recurrence within the flap itself, supporting the thesis that the fillet flap technique is oncologically safe [29].

However, in cases where FQA may be prevented or primary wound closure is possible, the overall prognosis of this special patient group must be taken into account, in particular when considering osteomusculocutaneous fillet flap reconstruction after FQA due to sarcoma. Even if primary tumor resection is successful, the disease-free five year survival of sarcoma patients requiring FQA is below 30% [53]. While Steinau et al. advocated that the use of osteomusculocutaneous fillet flaps even applies to a palliative reconstruction, the apparent limitations of ultra-radical interdisciplinary oncological surgery, albeit technically feasible, have to be critically reflected [2,25]. This holds true, in particular, when considering previously reported long-term impairment of respiratory function and of quality of life in patients with chest wall resection, as well as the significantly increased depressive symptoms of family members that often need to be consulted in the course of the intensive care treatment of critically ill patients [25,54,55].

Therefore, we advocate that the indication for FQA has to be individually considered and carefully evaluated with each patient after case discussion in a specialized interdisciplinary tumor board (if applicable, i.e., if FQA is considered to treat an underlying malignancy). However, if this process concludes that FQA is the best treatment option, radical tumor resections and subsequent osteomusculocutaneous fillet flap reconstruction, including TMR (when manageable in limited additional operating time), should be the first choice of surgical treatment.

## 5. Conclusions

In case of FQA and the need for free flap reconstruction, we consider the osteomusculocutaneous free fillet flap as the first choice. It enables the reconstruction of the chest wall integrity, provides support for a prosthesis socket, and improves the appearance of the shoulder contour. When using this technique, the remaining anatomical structures of the thorax, the vascular supply, and the distal resection margin of the amputated upper extremity are crucial components that have to be considered when the overall design of the flap is determined. If the inclusion of the elbow joint into the flap is not possible due to local tumor expansion or trauma, we recommend the use of the forearm and hand, as described. In general, we advocate for the additional implementation of TMR, as it can be performed quickly and is likely to reduce the occurrence of phantom limb and neuroma pain. However, the reviewed case series, as well as our own experience, emphasize that patients require careful evaluation of the benefit of FQA, as well as an individual solution for reconstructive surgery if the procedure is deemed to be the best possible option.

## Figures and Tables

**Figure 1 jpm-12-00560-f001:**
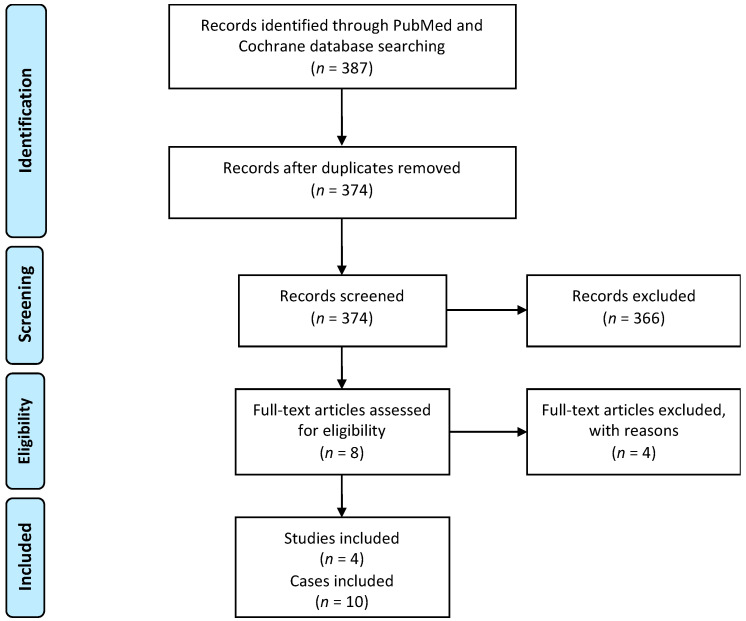
Study-selection algorithm based on the Preferred Reporting Systems for Systematic Reviews and Meta-Analysis (PRISMA) statement.

**Figure 2 jpm-12-00560-f002:**
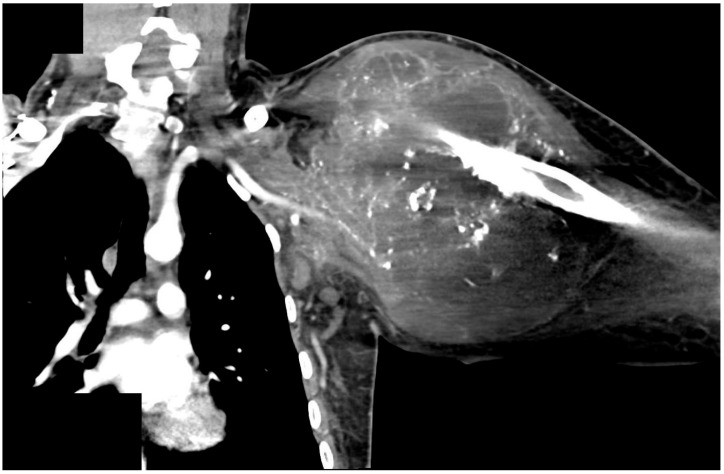
CT-Scan of a 25 year old male who presented with a chondroblastic osteosarcoma of the left proximal humerus, infiltrating the left glenohumeral joint and the muscles of the upper arm and rotator cuff, including latissimus dorsi and both pectoral muscles (staged at cT2 cN0, and cM1).

**Figure 3 jpm-12-00560-f003:**
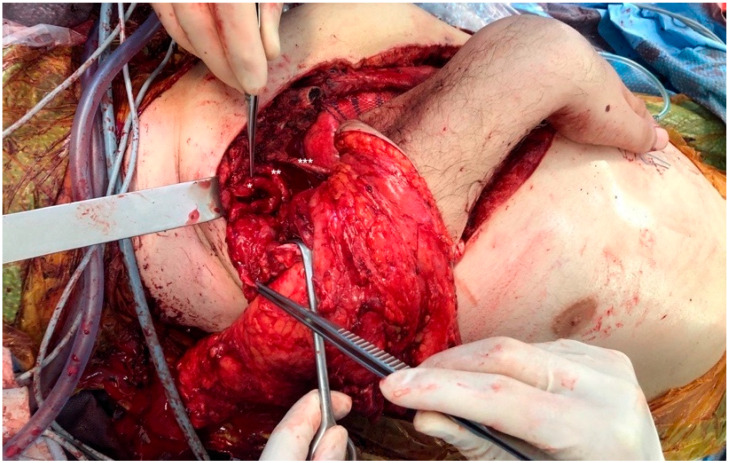
Intraoperative view of osteomusculocutaneous free fillet flap for defect coverage after forequarter amputation (FQA). The radius (and ulna) was shortened and attached to the sternum with a plate. Microsurgical anastomoses were performed between the brachial artery and thoracoacromial artery, and between the cephalic vein and the remaining stump of the subclavian vein. Subsequently, targeted muscle reinnervation (TMR) by epineural coaptation of the three forearm motor-nerves was performed: the superior trunk was connected to the median nerve (*), the middle trunk to the radial nerve (**), and the inferior trunk to the ulnar nerve (***).

**Figure 4 jpm-12-00560-f004:**
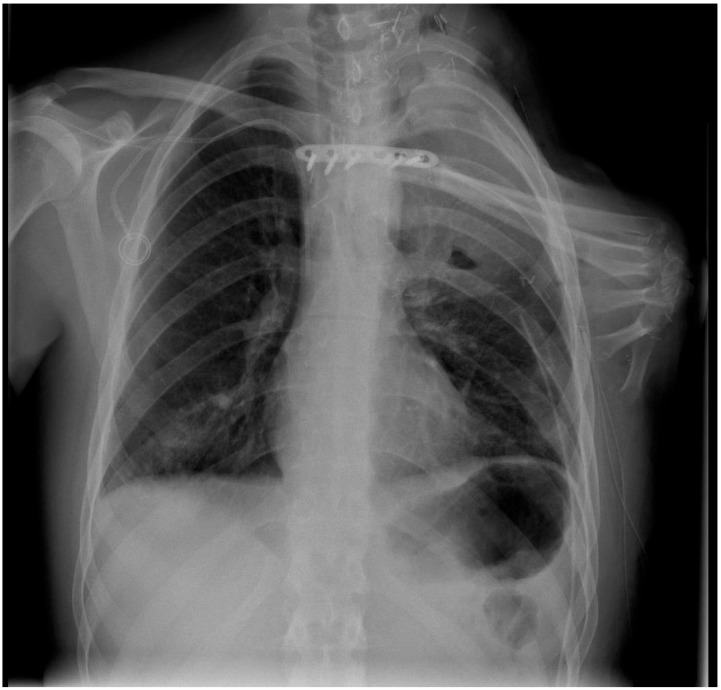
Osteomusculocutaneous free fillet flap including the tumor-free forearm for defect coverage and shoulder contour reconstruction (radiograph taken one week after surgery). The 90° flexed wrist, as well as the carpal and metacarpal bones, were incorporated into the flap to create a shoulder contour that would function as a prosthetic socket. Plate osteosynthesis was used to attach the sternum to the radius.

**Figure 5 jpm-12-00560-f005:**
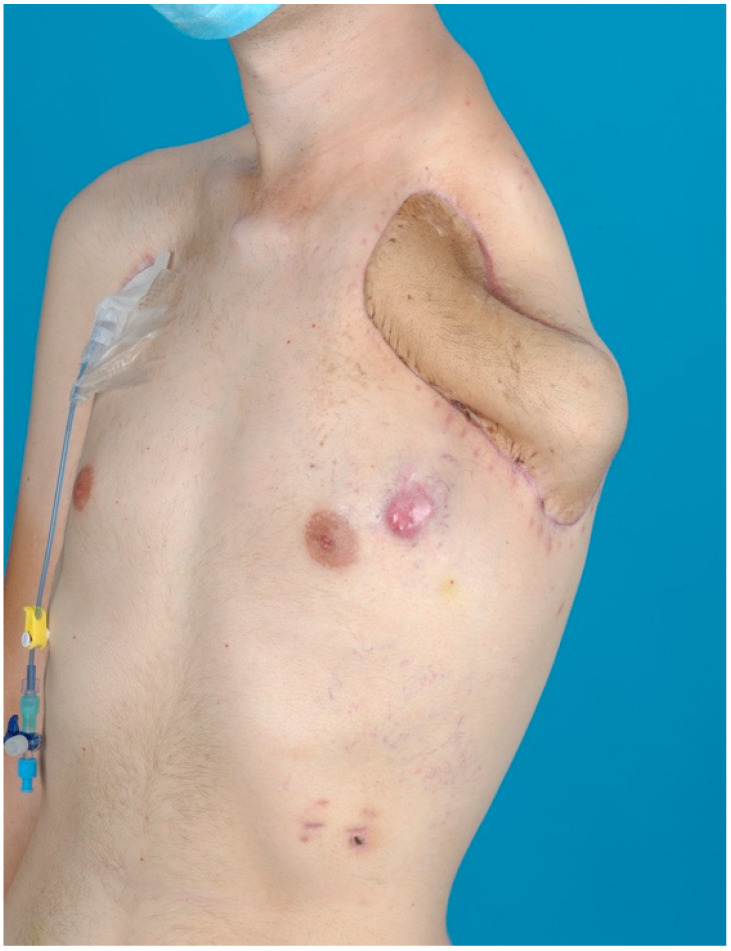
Clinical follow-up six weeks after osteomusculocutaneous free fillet flap defect coverage, including targeted muscle reinnervation (TMR). The results demonstrate an adequate reconstruction of the chest wall integrity, as a well as an improved appearance of the shoulder contour. The patient did not suffer from phantom limb or neuroma pain, and neurological examination showed an increasing tactile sensation of the flap. Due to disease progression in the course of treatment, cutaneous metastasis developed above the left breast.

**Table 1 jpm-12-00560-t001:** Overview of all 10 cases that previously described free osteomusculocutaneous fillet flaps for thoracic wall stabilization and shoulder contour reconstruction after FQA.

Ref.	Indication	Anastomosis	Reconstruction	Outcome
Steinau et al. (1992) [2]	46 year old male with 8th local recurrence of a chondrosarcoma (T3 N0 M0 G2) with infiltration of the brachial plexus and the thoracic wall. Palliative FQA with resection of ¾ of ribs 1–5 and partial removal of the sternum	Brachial artery to subclavian artery; brachial and superficial vein to the bifurcation of the external jugular vein	Fixation of radius and ulna with interosseous wires to remaining parts of the sternum and the sixth rib for thoracic wall stabilization	Exitus letalis 13 months after surgery due to bilateral pulmonary metastases
	22 year old male, recurrence of osteosarcoma (T3 N1 M0 G3), palliative FQA	Brachial artery to subclavian artery; brachial and superficial vein to the bifurcation of the external jugular vein	Radius and ulna were attached to the sternum and the thoracic wall with K-wires and strong circumferential wires	Revision due to an infected hematoma. Development of bilateral pulmonary metastases two months after surgery
	30 year old male, traumatic interscapulothoracic avulsion accident	Brachial artery to subclavian artery; brachial vein to subclavian vein and a superficial vein to the external jugular vein	Fixation of the Olecranon to the stump of the clavicle and the radius and ulna to the thoracic wall. Both with K-wires	No complications; wears a passive prosthetic replacement
Kuhn et al. (1994) [16]	21 year old male with an extensive recurrent desmoid tumor involving the chest wall from the clavicle to the 8th rib. Extensive FQA including ipsilateral hemithoracectomy and pneumectomy	Brachial artery to subclavian artery; cephalic vein to. internal jugular vein and basilic vein to innominate vein	Free forearm fillet flap with attachment of the ulna to the 2nd and 9th rib with screws and miniplates. The radius was removed completely	No complications; returned to work three months after surgery
Osanai et al. (2005) [17]	16 year old male with osteosarcoma, palliative FQA	Brachial artery to subclavian artery; brachial vein to subclavian vein	Plate osteosynthesis between the humerus and clavicle, 90° flexed elbow for shoulder contour reconstruction	Exitus letalis six months after surgery due to multiple pulmonary metastases
	56 year old female, primary malignant cystosarcoma phyllodes of the breast with local progression, extensive FQA including chest wall and rib resection (ribs 2 to 4)	Brachial artery to suprascapular artery; brachial vein to suprascapular vein	Insertion of the end of the clavicle into the enlarged marrow cavity of the humerus and fixation with nonabsorbable sutures, 90° flexed elbow for shoulder contour reconstruction	No evidence of local recurrence 10 months after surgery
Koulaxouzidis et al. (2014) [18]	46 year old male, traumatic FQA	Brachial artery to subclavian artery; Cubital vein to subclavian vein	Plate osteosynthesis between humerus and clavicle, 90° flexed elbow for shoulder contour reconstruction	Partial necrosis, three revision surgeries and split-thickness skin grafts
	59 year old female, radiation induced soft tissue sarcoma (pT2a, N0, M0, G3) with infiltration of the brachial plexus and ulceration, extended FQA including the lateral third of the clavicle	Brachial artery to subclavian artery; cubital vein to subclavian vein	Cerclage wire osteosynthesis of the humerus to the middle third of the clavicle, 90° flexed elbow for shoulder contour reconstruction	Three revision surgeries due to arterial thrombosis, wound dehiscence, and partial necrosis of the flap. No local recurrence or metastasis in two-year follow up
	73 year old female, radiogenic sarcoma with invasion of the brachial and cervical plexus, the scapula, lateral clavicle, first three ribs and the apex of the lung, extended FQA including resection of the first three ribs and lung apex	Brachial artery to internal thoracic artery; brachial vein to internal thoracic vein	Cerclage wire osteosynthesis of the humerus to the middle third of the clavicle, 90° flexed elbow for shoulder contour reconstruction	No complications; the patient died 14 years after surgery from a sarcoma-unrelated causes
	57 year old female, loco-regional persistence of an infiltrating lobular carcinoma of the breast 16 years after initial diagnosis and therapy. FQA was necessary due to infiltration of the brachial plexus and stenosis of the brachial vessels, infiltration of the biceps, triceps, and infraspinatus muscle as well as the scapula	Brachial artery to subclavian artery; cephalic vein to subclavian vein and brachial vein to external jugular vein	Plate osteosynthesis between humerus and clavicle, 90° flexed elbow for shoulder contour reconstruction	R1 resection, leading to re-excision with intraoperative radiation. Loco-regional recurrence after six years requiring another re-excision and adjuvant chemotherapy. Again, four years later the patient presented with cervical lymph node metastases leading to neck dissection. Subsequently, one year later, tumor recurrence at the thoracic wall

**Table 2 jpm-12-00560-t002:** Overview of the presented case using an osteomusculocutaneous fillet flap for defect coverage after FQA.

Indication	Anastomosis	Reconstruction	TMR	Outcome
25 year old male with central chondroblastic osteosarcoma (cT2 cN0, cM1), extended FQA, including resection of the clavicle and the first three ribs	Brachial artery to thoracoacromial artery and cephalic vein to subclavian vein	Plate osteosynthesis between radius and sternum, 90° flexed wrist and fixation sutures between metacarpals and the lateral thoracic wall	Nerve coaptation between superior trunk and median nerve, middle trunk and radial nerve, and inferior trunk and ulnar nerve	Discharged after 11 days, stable osseous framework, Exitus letalis three months after surgery due to disseminated, primarily pulmonal, metastases

## Data Availability

Not applicable.

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
