# Peer review of "Defect Coverage after Forequarter Amputation—A Systematic Review Assessing Different Surgical Approaches"

_jpm, 2022, doi:10.3390/jpm12040560_

Round 1

Reviewer 1 Report

The authors provide a review of the literature regarding "Defect Coverage after Forequarter Amputation". Overall the manuscript is well written and informative. Additionally they present a own case of this surgical procedure.

Overall, I believe the manuscript is fine in its current form. A short description to alternate reconstructive methods for this scenarios as well as their pros and cons would help

Author Response

Dear Sir or Madam,

we thank you for your time reviewing our work and also for considering our manuscript for publication in the Journal of Personalized Medicine. Moreover, we find your suggested improvements highly beneficial to the overall quality of the manuscript and are happy to implement these.

Point 1: Overall, I believe the manuscript is fine in its current form. A short description to alternate reconstructive methods for this scenarios as well as their pros and cons would help

Response 1: We added a short description and evaluation of alternate reconstructive methods in the 2nd paragraph of the discussion (line 231-245). Here, we intentionally refrained from a too detailed analysis as we intend to keep the focus on the core topic of the review. However, we are happy to further extend this paragraph, if you believe a more thorough analysis should be included.

Sincerely,

Nikolaus Wachtel

Reviewer 2 Report

It is a well written paper on a rare case. The conclusion on trageted reinnervation, however, is not substantial and cannot be concluded from this limited experience. The patient survived only 3 months. The authors just believe. additionally, the "review" does not add any substantial information.

In fig. 5. there is apparently a cutaneous metastasis which is not mentioned in the legend.

Author Response

Dear Sir or Madam,

we thank you for your time reviewing our work and also for considering a possible publication of our manuscript in the Journal of Personalized Medicine.

Point 1: It is a well written paper on a rare case. The conclusion on trageted reinnervation, however, is not substantial and cannot be concluded from this limited experience. The patient survived only 3 months. The authors just believe. additionally, the "review" does not add any substantial information.

Response 1: Indeed, our paper discusses a case, which is rarely presented to the surgeon in such extremity in the clinical setting.  Nevertheless, we believe that, in particular, larger hospitals, with an established department for plastic surgery and a close interdisciplinary approach to soft tissue and bone tumors, experience similar cases on a rare but regular basis. Similarly, during the planning of the presented procedure we were surprised that, even after an extensive literature research, only a sparse amount of literature is available on this topic. We therefore believe that a thorough analysis of the surgical technique contributes positively to the field, in particular with regard to the focus of the current special issue.

Moreover, while your analysis of the limited validity of our results with regard to conclusive evidence of the benefits of TMR is undeniable correct, we do see a high benefit to cost ratio of this procedure. We therefore decided to present our findings on the feasibility of TMR as well as the possibility of functional improvement of phantom pain management when performing this procedure in combination with an epaulette flap. However, as rightfully pointed out by your assessment of our work, this is not conclusive evidence of a possible benefit of TMR. We therefore implemented several changes in the manuscript to clarify the suggestive nature of our conclusions.

Point 2: In fig. 5. there is apparently a cutaneous metastasis which is not mentioned in the legend.

Response 2: We added the correct description of the cutaneous metastasis to the legend of Figure 5.

Sincerely,

Nikolaus Wachtel

Round 2

Reviewer 2 Report

While I can see the changes to legend of fig. 5 I cannot identify the changes made to the text. The changes are not highlighted. The authors have to do that to allow the reviewers to see the changes. no news here?!

Author Response

Please refer to the comments made in Review Report (Round 3).

Round 3

Reviewer 2 Report

I just asked for a reevaluation of the targeted reinnervation. The authors wrote: "We therefore implemented several changes in the manuscript to clarify the suggestive nature of our conclusions."

However, there is no modification on this topic. Furthermore, the authors wrote in their letter that they "believe". I am sorry but if they are not able to change such a simple ask I think the paper has to be rejected.

Author Response

Dear Sir or Madam,

Point 1: While I can see the changes to legend of fig. 5 I cannot identify the changes made to the text. The changes are not highlighted. The authors have to do that to allow the reviewers to see the changes. no news here?!

Response 1: We apologize for the inconvenience. Normally, we manually highlight the changes made. However, for this manuscript we used the “Track Changes” function of MS Word as demanded by the MDPI Editorial office. Obviously there is a problem with the tracking of changes after upload of the manuscript. We therefore additionally highlighted the changes in the newest version to ensure their visibility. We truly hope that this problem is solved now.

Point 2: I just asked for a reevaluation of the targeted reinnervation. The authors wrote: "We therefore implemented several changes in the manuscript to clarify the suggestive nature of our conclusions."

However, there is no modification on this topic. Furthermore, the authors wrote in their letter that they "believe". I am sorry but if they are not able to change such a simple ask I think the paper has to be rejected.

Response 2: Thank you very much for the clarification. We reevaluated the argument for TMR with regard to several recent clinical studies. Thus, several authors demonstrated the high effectiveness of TMR with regard to prevention and treatment of chronic pain in major limb amputees (also when compared to the gold standard for neuroma treatment). The inclusion of this procedure in fillet-flap reconstruction after FQA is therefore highly likely to be beneficial for these patients. Please also see line 305 to 319 for a detailed discussion of the relevant studies. Nevertheless, due to the single case experience and limited follow-up of the presented case we clarified the suggestive nature of this conclusion as rightfully suggested by you (line 303, 305, and 336).

Again, we apologize for the inconvenience and would be happy to further revise the manuscript.

Sincerely,

Nikolaus Wachtel